# Plasma Cytokines/Chemokines as Predictive Biomarkers for Lymphedema in Breast Cancer Patients

**DOI:** 10.3390/cancers15030676

**Published:** 2023-01-21

**Authors:** Anna R. Vang, Simona F. Shaitelman, John C. Rasmussen, Wenyaw Chan, Eva M. Sevick-Muraca, Melissa B. Aldrich

**Affiliations:** 1UT Brown Foundation Institute of Molecular Medicine for the Prevention of Human Diseases, Houston, TX 77030, USA; 2The University of Texas MD Anderson Cancer Center UTHealth Houston Graduate School of Biomedical Sciences, Houston, TX 77030, USA; 3MD Anderson Cancer Center, Houston, TX 77030, USA

**Keywords:** lymphedema, breast cancer, cytokine/chemokine, biomarkers, dermal backflow, NIRF-LI

## Abstract

**Simple Summary:**

Lymphedema (LE) is characterized by arm, leg, trunk, or head/neck swelling, pain, depression, and cellulitis risk. Approximately 40% of breast cancer patients develop LE. Early LE diagnosis and treatment improve outcomes and minimize costs. As there is currently no cure, identifying predictive markers for breast cancer-related lymphedema (BCRL) could immensely reduce the financial burden on these patients and increase treatment success. This study identifies plasma cytokines/chemokines that predict BCRL development over a year before clinically recognized symptoms appear.

**Abstract:**

Breast cancer-related lymphedema (BCRL) occurs in ~ 40% of patients after axillary lymph node dissection (ALND), radiation therapy (RT), or chemotherapy. First-line palliative treatment utilizes compression garments and specialized massage. Reparative microsurgeries have emerged as a second-line treatment, yet both compression and surgical therapy are most effective at early stages of LE development. Identifying patients at the highest risk for BCRL would allow earlier, more effective treatment. Perometric arm volume measurements, near-infrared fluorescent lymphatic imaging (NIRF-LI) data, and blood were collected between 2016 and 2021 for 40 study subjects undergoing treatment for breast cancer. Plasma samples were evaluated using MILLIPLEX human cytokine/chemokine panels at pre-ALND and at 12 months post-RT. A Mann–Whitney *t*-test showed that G-CSF, GM-CSF, IFN-2α, IL-10, IL-12p40, IL-15, IL-17A, IL-1β, IL-2, IL-3, IL-6, and MIP-1β were significantly higher at pre-ALND in those presenting with BCRL at 12 months post-RT. MIP-1β and IL-6 were significantly higher at pre-ALND in those who developed dermal backflow, but no BCRL, at 12 months post-RT. Plasma IL-15, IL-3, and MIP-1β were elevated at 12 months after RT in those with clinical BCRL. These findings establish BCRL as a perpetual inflammatory disorder, and suggest the use of plasma cytokine/chemokine levels to predict those at highest risk.

## 1. Introduction

Breast cancer-related lymphedema (BCRL) is characterized by the accumulation of stagnant lymph, subdermal fat/adipose tissue, and skin fibrosis in the upper extremities and trunk after radiation, chemotherapy, and/or lymph node dissection [1,2,3]. BCRL affects approximately 40% of breast cancer survivors [4,5,6,7,8]. As there is currently no cure [9], the management of lymphedema (LE) strives to improve quality of life (QOL). The primary treatment for LE is the use of compression to help reduce swelling and maintain arm volume reduction. In the first two weeks after LE diagnosis, patients are recommended to undergo one hour of manual lymphatic drainage (MLD) therapeutic massage daily to remove stagnant lymph, followed by immediate bandage wrapping to maintain the decreased arm volume. After maximal arm volume reduction, compression sleeves are prescribed for 24/7 wear to maintain arm volume and prevent subdermal adipose tissue buildup. If MLD fails, reparative lymphatic microsurgeries, such as vascularized lymph node transplant (VLNT) and lymphovenous bypass (LVB), can improve outcomes. VLNT serves to transplant a healthy/functional lymph node flap from an unaffected part of the body to an affected area. LVB redirects the flow of lymph by connecting affected lymphatic vessels to draining adjacent veins. These microsurgeries reduce arm swelling by 30% on average, with the continued use of compression garments [10]. The lymphatic microsurgical preventative healing approach (LYMPHA), a technique that is increasingly being adopted, creates a shunt between a lymphatic channel and a draining blood vessel at the time of ALND. One study showed that LYMPHA reduced the incidence of LE from 40% to 12.5% [11]. Treatment for LE is expensive and not fully covered by insurance.

BCRL’s molecular etiology is not completely known, but one study showed that stagnant lymph provides free fatty acids (FFAs) that signal subdermal adipose cells to grow and divide [12]; skin fibrosis follows, and cellulitis risk increases. Cellulitis results in trophic skin changes that can progress to sepsis if not treated or detected early, requiring hospitalization. LE also takes a toll on psychological health factors, as nearly half of breast cancer survivors have reported some level of LE-related distress and depression, anxiety, fatigue, and inability to participate in social activities [13,14].

Arm volume can be assessed by several means, including tape measurement, water displacement, or perometer measurement. Perometry determines the relative volume change (RVC) between the affected and unaffected arms. Patients with RVC scores ≥ 5% are diagnosed with LE defined by the International Society of Lymphology (ISL) [15,16]. Variable criteria for arm volume increase complicates uniform BCRL diagnosis. Breast cancer patients are not usually assessed for LE until at least 3 months after oncologic treatment, to allow for the resolution of swelling due to surgery or RT [15,17]. 

“See through the skin” near-infrared fluorescence lymphatic imaging (NIRF-LI), however, has recently shown that the dermal backflow of lymph, a hallmark of BCRL, is present 8–23 months before arm swelling is evident [17]. Dermal backflow most likely results after lymphatic pumping failure, which we and others have shown can result from inflammatory cytokine actions on nitric oxide levels that can interfere with normal nitric oxide fluctuations that drive lymphatic pumping (Figure 1) [18]. In addition to the imaging surveillance of lymphatics, biomarkers of failing lymphatic function, such as inflammatory cytokines/chemokines, could aid the early identification of breast cancer patients most at risk for developing BCRL [19,20,21].

We hypothesize that elevated plasma cytokine and chemokine levels precede BCRL development, and thus, could identify those at highest risk much earlier than arm volume increase. Biomarkers of BCRL risk could allow targeted surveillance and early intervention, improving patient QOL and lowering BCRL-associated medical costs.

## 2. Materials and Methods

### 2.1. Study Subjects and Design

Breast cancer patients of at least 18 years of age with no prior radiation therapy targeted to lymph nodes, who were scheduled to undergo treatment—including mastectomy or breast-conserving surgery with ALND and radiation therapy—at the University of Texas MD Anderson Cancer Center (MDACC), were recruited for the study. Breast cancer patients with additional underlying chronic illness or disease, known/suspected iodine allergy, breastfeeding, pregnancy, or an inability to keep still for a one-hour image session, were excluded from the study. The full inclusion and exclusion criteria are listed in the Appendix A.

Of the 80 study subjects who consented, 40 were excluded from our analysis—seven passed away from distant metastases before completing the study, 11 dropped out, one developed locally-regionally recurrent breast cancer, and 21 missed multiple or analysis-relevant visits due to SARS-CoV-2. The demographics of the 40 study subjects are shown in Table 1.

Observation of each study subject was conducted over 18 months with data collection at six specified time points: pre-ALND, post-ALND, end of RT, and at 6-, 12-, and 18-months post-RT. Blood specimens and perometric scores were collected at all time points, while NIRF-LI images were collected at only five time points (Figure 2). Data from four weeks post-ALND, end of RT, and six months post-RT were excluded from this study due to insufficient numbers of subjects developing clinical LE at those time points. Data from 18 months post-RT were not used due to COVID-19 interruptions that prevented us from obtaining sufficient samples.

All subjects signed their informed consent. The study was conducted under the Declaration of Helsinki and approvals from the Committees for Protection of Human Subjects/Institutional Review Boards (2016-0170 and HSC-15-1021, respectively) at both MD Anderson Cancer Center and The University of Texas Health Science Center, as well as FDA combinational Investigation New Drug application 106,345 for off-label administration and the use of ICG with NIRF-LI (NCT 02949726).

### 2.2. Near-Infrared Fluorescent Lymphatic Imaging (NIRF-LI)

The study subjects were imaged at five time points: pre-ALND, post-ALND, and at 6-, 12-, and 18-months post-RT. A total of eight intradermal injections of 0.1 cm^3^/25 µg of indocyanine green (ICG) were made into the dorsal hand and ventral wrist areas of affected and unaffected arms at each visit, for a total dose of 200 µg. Real-time lymphatic images of the dorsal and ventral upper extremities and the axillary were obtained for each arm, for five to ten minutes per view, for a total of 30–45 min. Extremities were illuminated with 785 nm of excitation light, and the emitted fluorescence was captured by a custom, 16-bit, frame transfer, charge-coupled device camera at a field of view of 350–1900 cm^2^ [22]. Acquired fluorescent images from each session were processed into a stacked video file using ImageJ software (ImageJ version 1.2.4, RRID: SCR_003070) and analyzed. NIRF-LI -visible extravascular dye/dermal backflow was measured as described below.

### 2.3. Perometric (RVC) Arm Volume Measurement/Clinical Diagnosis of LE

A horizontal Perometer 400NT (Perosystem) was used to measure the RVC of each study subject at every visit. Volumetric arm measurements were calculated from the average of three perometer measurements of each arm. The formula used was RVC = (A_2_U_1_)/(U_2_A_1_) − 1, where A represents the arm volumes on the ipsilateral (affected) arm and U represents the arm volumes of the contralateral (unaffected) arm [23]. A_1_ and U_1_ are baseline arm volume measurements, and A_2_ and U_2_ are the follow-up arm volume measurements. Patients with RVC values ≥ 5% were diagnosed with clinical LE [15,16].

### 2.4. Extra Vascular Dye (EVD) or Dermal Backflow


(1)
$${\mathrm{BSA}\;m}^{2}=\sqrt{\frac{\left(\mathrm{Height}\;\mathrm{cm}\;\times \mathrm{weight}\;\mathrm{kg}\right)}{3600}}\;$$



(2)
$$\%\;\mathrm{EVD}=\left(\frac{\left(\mathrm{Total}\;\mathrm{affetced}\;\mathrm{EVD}\;{\mathrm{cm}}^{2}/10,000\right)}{\left({\mathrm{BSAcm}}^{2}\times (0.09\;\mathrm{or}\;0.075\right)}\right)\times 100$$


Using the height and weight of each subject, body surface area (BSA) was calculated (1). A single arm surface area (2), per Wallace Rule of Nines [24], was calculated as 9% of BSA for subjects with body mass index (BMI) ≤ 33, and 7.5% for subjects with BMI > 33. The percent of arm surface area displaying dermal backflow was determined by calculating the dermal backflow-affected area of the arm and dividing by the arm surface area, by comparing the total pixels in the area of a paper rectangular grid of an area 6.0 cm^2^ (2.0 × 3.0 cm) with the total pixels in the area of dermal backflow observed in a still NIRF-LI image, at identical distances from the NIRF-LI camera lens. If %EVD was over 1%, we considered the images positive for dermal backflow.

### 2.5. Blood Plasma Isolation

Whole blood tubes with EDTA were centrifuged at 2000× *g* for 20 min at four degrees Celsius. Plasma was isolated and aliquoted (170–175 µL) into microtubes before storage at −80 degrees.

### 2.6. MILLIPLEX Map Human Cytokine/Chemokine Magnetic Bead Panel

The plasma samples were analyzed using Human Cytokine/Chemokine/Growth Factor Panel A Magnetic Bead Panel 96-well plate assay purchased from Millipore Sigma (St. Louis, MI, USA) (catalog #: HCYTA-60K). A total of 14 cytokines/chemokines were run for each plasma sample: G-CSF, GM-CSF, IL-12p40, IFN-α2, IL-10, IL-15, IL-17A, IL-1β, IL-2, IL-3, IL-6, IP10, MIP-1β, TNF-α. The plasma samples were processed using the protocol provided by the manufacturer. Plasmas from three normal healthy controls were included in the plates for verification but not used in statistical analysis.

### 2.7. Statistical Analysis

Outliers for each data set were calculated using the quartile functions in Excel. Pearson correlation values were calculated for %RVC and %EVD with cytokine/chemokine levels. Statistical significance was determined using the Mann–Whitney unpaired nonparametric *t*-test and Wilcoxon paired non-parametric *t*-test calculated from GraphPad Prism version 9.00. *p*-values < 0.05 were deemed significant. Correlation values between 0.00 and 0.10 were considered negligible, between 0.10 and 0.39 were weak, between 0.40 and 0.69 were moderate, between 0.70 and 0.89 were strong, and between 0.90 and 1.00 were very strong [25]. Subjects who did not develop BCRL and/or dermal backflow 12 months post-RT were used as controls. For each data set, 0–2 outliers were removed if they were outside of the interquartile range.

## 3. Results

### 3.1. Pre-ALND Cytokine/Chemokine Levels in Patients Who Developed BCRL 12-Months Post-RT Were Elevated

Pre-ALND cytokine/chemokine levels for subjects diagnosed with clinical BCRL (≥5% RVC) at 12-months post-RT were compared to subjects who did not develop clinical LE at 12-months post-RT. G-CSF, GM-CSF, IFN-α2, IL-10, IL-12p40, IL-15, IL-17A, IL-1β, IL-2, IL-3, IL-6, and MIP-1β were significantly higher at pre-ALND in those who developed LE at 12-months post-RT compared to those who did not develop BCRL 12-months post-RT (Figure 3).

### 3.2. Several Plasma Cytokine/Chemokine Levels Were Elevated at 12 Months after RT in Those with Clinical BCRL

Plasma IL-15, IL-3, and MIP-1β levels were found to be significantly higher at 12-months post-RT in comparison to those who did not display BCRL at 12-months post-RT (Figure 4). Other plasma cytokine/chemokine levels did not reach significance, but several trended upwards (Appendix A).

### 3.3. Subjects Displaying Dermal Backflow One Year after RT Showed Elevated Cytokine/Chemokine Levels at Pre-ALND

Cytokine/chemokine levels at pre-ALND were analyzed, comparing those who displayed dermal backflow at 12-months post-RT to those who did not. Pre-ALND plasma IL-6 and MIP-1β were significantly higher in those who developed dermal backflow at 12-months post-RT compared to those who did not (Figure 5). Other plasma cytokine/chemokine levels did not reach significance, but many trended upwards (Appendix A).

### 3.4. Several 12-Months Post-RT Plasma Cytokine/Chemokine Levels Trended Higher in Subjects with Dermal Backflow at 12-Months Post-RT

Most cytokine/chemokine levels at 12-months post-RT were not significantly higher for those subjects with backflow at 12-months post-RT (Appendix A). Although statistically insignificant, IFN-α2, IL-12p40, IL-15, MIP-1β, and TNF-α levels trended higher in subjects who displayed dermal backflow at 12-months post-RT (Figure 6).

### 3.5. Several Cytokine/Chemokine Levels Were Elevated at Pre-ALND in Subjects with Both Dermal Backflow and Clinical BCRL/LE at 12-Months Post-RT

GM-CSF, IFN-α2, IL-12p40, IL-15, IL-6, TNF-α, and MIP-1β were significantly higher at the pre-ALND time point in those with both BCRL/LE and dermal backflow at 12-months post-RT in comparison to those who did not develop BCRL or dermal backflow 12-months post-RT (Figure 7). Other plasma cytokine/chemokine levels did not reach significance, but many trended upwards (Appendix A).

### 3.6. Several 12-Months Post-RT Cytokines/Chemokines Were Elevated in Subjects with Both Clinical BCRL and Dermal Backflow Compared to Subjects with Neither Clinical BCRL nor Dermal Backflow

IL-15, IL-3, and MIP-1β were found to be significantly higher in those who developed LE and dermal backflow at 12-months post-RT in comparison to those who did not develop LE or dermal backflow at 12-months post-RT (Figure 8). Other plasma cytokine/chemokine levels did not reach significance, but many trended upwards (Appendix A).

### 3.7. Pearson Correlation Coefficients Comparing %RVC and %EVD to Cytokine/Chemokine Levels Exhibited Negligible to Moderate Relevance

Pearson’s coefficient was used to determine the correlation between cytokine/chemokine levels to perometric scores and dermal backflow. The correlation values for all four comparison groups yielded values between negligible and moderate correlation (Table 2). Only IL-17A and IL-1β yielded a moderate correlation when comparing cytokine levels to perometric scores.

## 4. Discussion

At present, it is hard to predict who will develop BCRL. Current treatment methods only improve QOL, as there is no known cure [9]. Several studies have shown that early detection and treatment lead to better outcomes and health cost savings [17,26].

Our recent study found that dermal backflow, detected by NIRF-LI, is an imaging biomarker that can detect lymphatic dysfunction 8–23 months before breast cancer patients are clinically diagnosed with LE by perometry [17]. In the present study, we identified plasma cytokine/chemokine biomarkers that are predictive of BCRL development more than a year in advance of clinically diagnosed BCRL. These findings support our hypothesis that elevated plasma cytokine and chemokine levels precede BCRL development, therefore identifying those at risk before being clinically diagnosed.

We initially started with a 38 cytokine/chemokine panel, but after three trials, we found that only 14 were relevant to our study. The elevation of these 14 cytokines/chemokines was not consistent for all comparisons. For example, IL-15, IL-3, and MIP-1β were elevated at 12-months post-RT in those with clinical LE at 12-months post-RT, while only IL-15 and MIP-1β were elevated at 12-months post-RT in those with both clinical LE and dermal backflow at 12-months post-RT. Only IL-6 and MIP-1β were elevated at pre-ALND in those displaying dermal backflow at 12-months post-RT, while no cytokines were elevated at 12-months post-RT in those displaying dermal backflow at 12-months post-RT. Despite the lack of uniformity of elevated cytokine/chemokine levels at different time points, our analysis indicates a predictive value for many of these cytokines/chemokines as biomarkers and evidence of an ongoing inflammatory presence in BCRL.

The elevation of IFN-α2, IL-12p40, IL-15, MIP-1β, and TNF-α at 12-months post-RT in subjects displaying dermal backflow at 12-months post-RT, while not statistically significant, further supports the concept that BCRL is characterized by persistent inflammation and warrants further study. MIP-1β, IL-15, and IL-1β were either trending or significantly elevated in both the clinically diagnosed BCRL group and the dermal backflow group at both pre-ALND and 12-months post-RT. Of note, MIP-1β coerces cells to produce TNF-α, IL-1β, and IL-6, which studies have shown to dampen lymphatic pumping [18,27,28]. TNF-α levels, although never statistically significant, were consistently elevated in those displaying dermal backflow. IL-15 strongly associates with the defense and modulation of immune cells in both innate and adaptive immunity [29], and IL-1β mediates the release of other pro-inflammatory cytokines [30,31], suggesting a vicious cycle of inflammation associated with BCRL [32].

The Pearson correlation values for the perometry readings and cytokine/chemokine levels were likely affected by the use of compression once dermal backflow was detected. Arm volumes usually decrease with the use of MLD and compression, and may not reflect the cryptic dysfunctions seen with NIRF-LI.

The key limitations of this study include the small sample size and uncertainty of compression use. Despite recruiting 80 subjects, we lost half of them due to COVID-19, death, and/or other health complications, severely reducing our sample size. A larger sample size, with a burden analysis, using odds ratios and 95% confidence intervals derived with logistic regression, could be used to produce a BCRL predictive test kit similar to those used for breast cancer susceptibility genetic screening [33]. Plasma cytokine levels could be validated in a future study using MILLIPLEX kits produced by other manufacturers and individual ELISA kits. We could not directly gauge the effects of compression use once dermal backflow and/or BCRL were detected, as we had no way to monitor subject compression use. A future study measuring cytokine/chemokine levels after monitored compression use could verify the effects on inflammation.

## 5. Conclusions

Using cytokine/chemokine MILLIPLEX assays and NIRF-LI imaging, we found elevated cytokines/chemokines at baseline/pre-ALND to be predictive of who will develop BCRL and dermal backflow more than one year later. In addition, we found evidence of an ongoing cycle of inflammation associated with BCRL one year after RT, suggesting that BCRL is a systemic disease.

Cytokine screenings could be offered to patients at highest risk, particularly Black cancer patients, as well as inflammatory breast cancer (IBC) and triple-negative breast cancer patients [2,34,35]. In cases of positive screening results, early physiotherapy and/or reparative microsurgeries, as well as anti-inflammatory dietary, exercise, and pharmaceutical interventions, could be prescribed. Future studies could determine optimal timing and duration for such therapies. In summary, our findings establish BCRL as a perpetual inflammatory disorder and suggest the use of plasma cytokine/chemokine levels to predict those at highest risk.

## Figures and Tables

**Figure 1 cancers-15-00676-f001:**
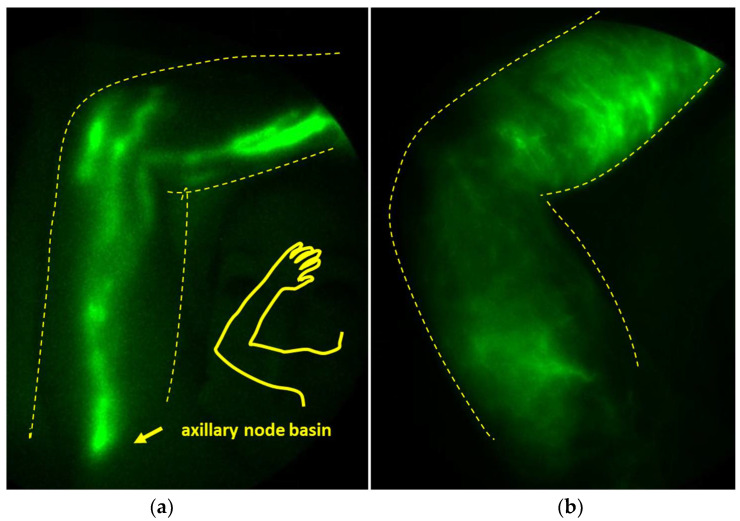
Near-infrared fluorescence lymphatic imaging (NIRF-LI) of the right upper-proximal extremity, including the axilla, depicting (**a**) a healthy lymphatic structure with visible lymphatic vessels and axillary node basin, (**b**) dermal backflow.

**Figure 2 cancers-15-00676-f002:**
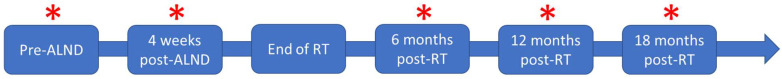
Longitudinal surveillance and data collection at pre-ALND, post-ALND, end of RT, and at 6-, 12-, and 18-months post-RT. Perometer arm measurements and blood samples were obtained at all visits. NIRF-LI image sessions were only conducted at pre-ALND, four weeks post-ALND, and at 6-, 12-, and 18-months post-RT (red asterisks).

**Figure 3 cancers-15-00676-f003:**
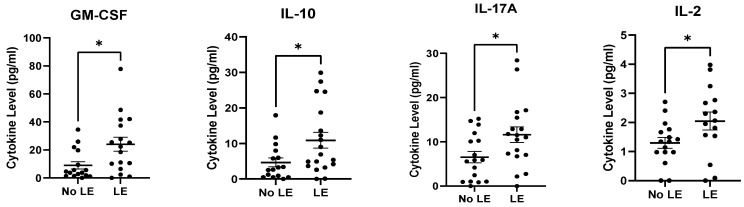
Pre-ALND plasma cytokine/chemokine levels were elevated in subjects displaying clinical BCRL/LE at 12-months post-RT. Of the 14 cytokines analyzed, only IP10 and TNF-α were not significantly elevated. * *p* ≤ 0.05, ** *p* ≤ 0.01, *** *p* ≤ 0.001, ns = *p* > 0.05.

**Figure 4 cancers-15-00676-f004:**
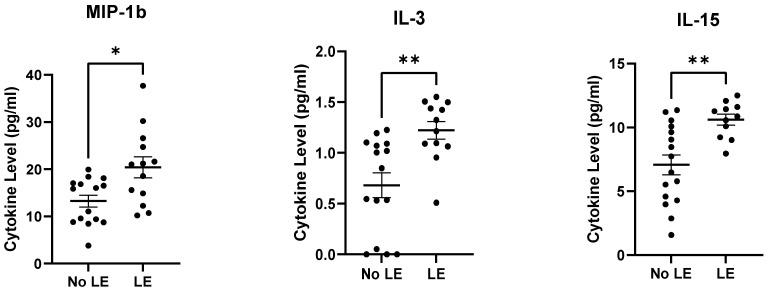
Plasmas IL-15, IL-3, and MIP-1β levels at 12-months post-RT were significantly elevated in subjects who were diagnosed with clinical BCRL at 12-months post-RT compared to those who did not. * *p* ≤ 0.05, ** *p* ≤ 0.01.

**Figure 5 cancers-15-00676-f005:**
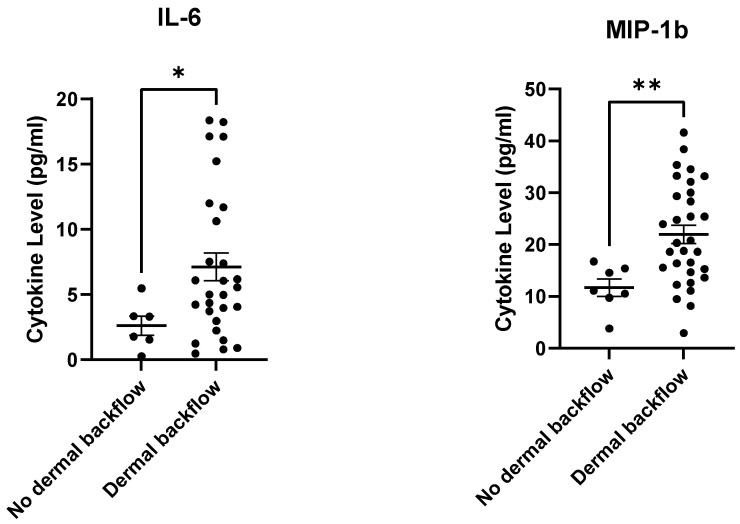
Pre-ALND IL-6 and MIP-1β plasma levels were higher in subjects displaying dermal backflow at 12-months post-RT. * *p* ≤ 0.05, ** *p* ≤ 0.01.

**Figure 6 cancers-15-00676-f006:**
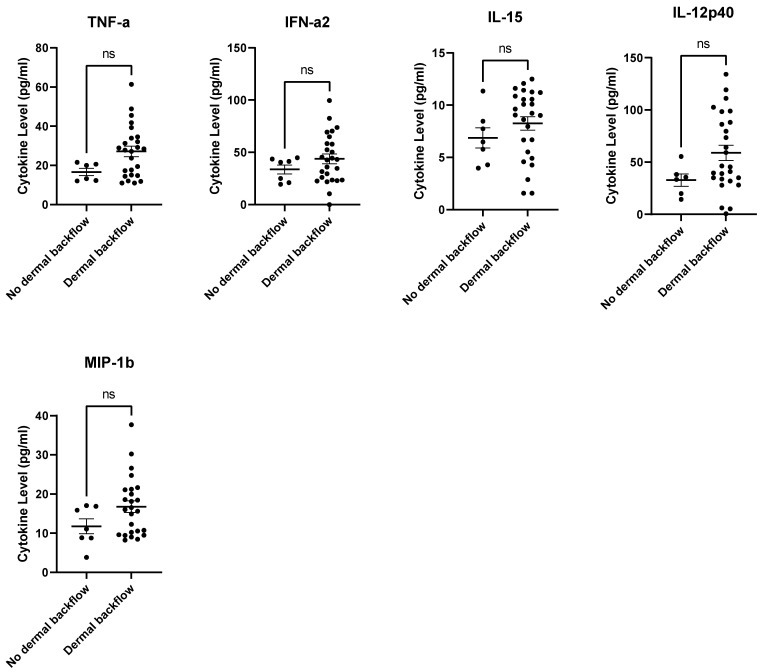
IFN-α2, IL-12p40, IL-15, MIP-1β, and TNF-α levels trended higher at 12-months post-RT in subjects with dermal backflow. (*p*-values = 0.2463, 0.1071, 0.2119, 0.1230, 0.0962, respectively). ns = *p* > 0.05.

**Figure 7 cancers-15-00676-f007:**
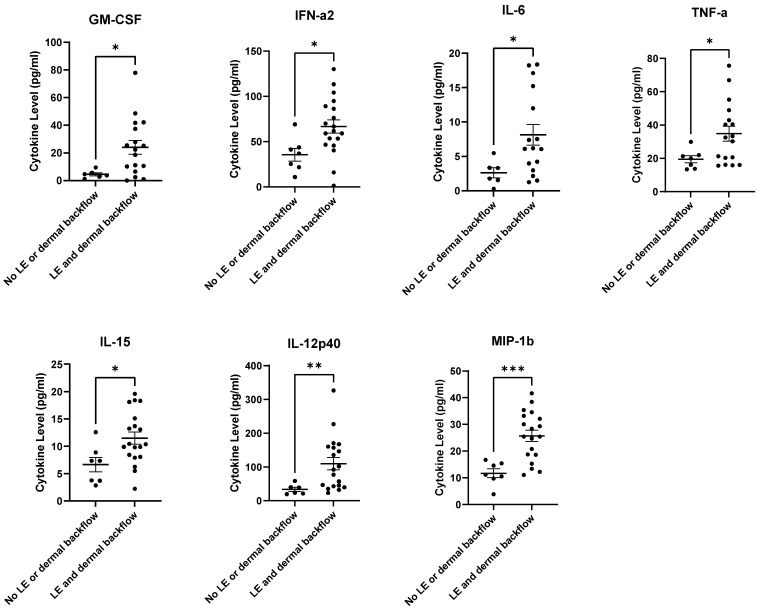
GM-CSF, IFN-α2, IL-12p40, IL-15, IL-6, TNF-α, and MIP-1β were significantly higher at pre-ALND in subjects with both clinical BCRL/LE and dermal backflow at 12-months post-RT compared to subjects with no clinical BCRL or dermal backflow. * *p* ≤ 0.05, ** *p* ≤ 0.01, *** *p* ≤ 0.001.

**Figure 8 cancers-15-00676-f008:**
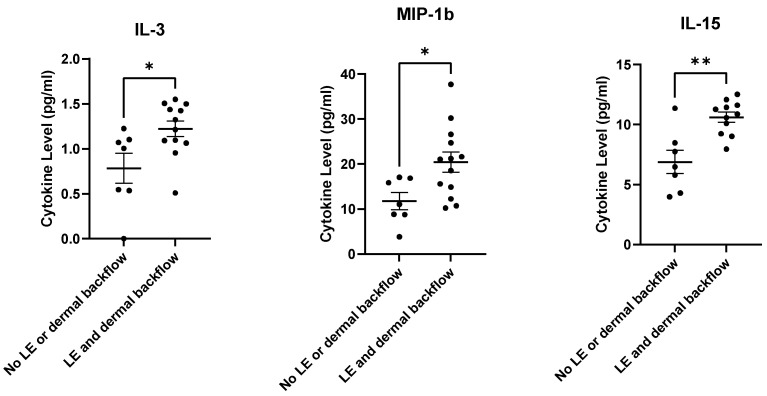
Plasma cytokine/chemokine levels at 12-months post-RT were elevated in those who had both clinical LE and dermal backflow at 12-months post-RT. * *p* ≤ 0.05, ** *p* ≤ 0.01.

**Table 1 cancers-15-00676-t001:** Study subject demographics.

Characteristics	Value
Age, year, median (range)	48.15 (26–68)
Race, *n* (%)	
Black	3 (7.5)
Other (Asian, American Indian/Alaska Native, multi-race)	5 (12.5)
White	32 (80)
Ethnicity, *n* (%)	
Hispanic or Latino	6 (15)
Non-Hispanic	34 (85)
Sex, *n* (%)	
Female	40 (100)
Male	0 (0)
Body mass index, mean (range), *n* (%)	
Underweight (<18.5)	1 (2.5)
Normal weight (18.5–24.9)	10 (25)
Overweight (25.0–29.9)	13 (32.5)
Obese (≥30.0)	16 (40)
Clinical T category, *n* (%)	
Tx	1 (2.5)
T1	4 (10)
T2	15 (37.5)
T3	10 (25)
T4b	4 (10)
T4d	6 (15)
Clinical N category, *n* (%)	
N1	16 (40)
N2	4 (10)
N3a	7 (17.5)
N3b	2 (5)
N3c	11 (27.5)
Neoadjuvant chemotherapy, *n* (%)	38 (95)
Taxanes, *n* (%)	37 (92.5)
Anthracyclines, *n* (%)	34 (85)
Number of lymph nodes removed at ALND, median (range)	23.37 (6–39)
Number of lymph nodes involved at ALND, median (range)	4.57 (0–36)
Lymphovascular space invasion, *n* (%)	10 (25)
Extracapsular extension, *n* (%)	13 (32.5)
Lumpectomy, *n* (%)	10 (25)
Mastectomy, *n* (%)	30 (75)
Cumulative radiation dose, Gy, median	49.88
Total number of fractions of radiation, median	26

**Table 2 cancers-15-00676-t002:** Pearson correlation values comparing pre-ALND and 12-months post-RT plasma cytokines/chemokine levels to %RVC and %EVD.

Cytokine/Chemokine	R^2^ for Pre-ALND pg/mL and RVC at 12-Months Post-RT	R^2^ for 12-Months Post-RT pg/mL and RVC at 12-Months Post-RT	R^2^ for Pre-ALND pg/mL and %EVD at 12-Months Post-RT	R^2^ for 12-Months Post-RT pg/mL and %EVD at 12-Months Post-RT
G-CSF	0.0362	0.0079	0.0005	0.0066
GM-CSF	0.1189 *	0.2475 *	0.0026	0.0344
IFN-α2	0.0025	0.3862 *	0.0449	0.0256
IL-10	0.0089	0.0107	0.0166	0.0454
IL-12p40	0.0124	0.0818	0.0164	0.1437 *
IL-15	0.0248	0.003	0.00002	0.1563 *
IL-17A	0.0384	0.5207 **	0.078	0.0557
IL-1β	0.1636 *	0.4829 **	0.0006	0.0296
IL-2	0.0965	0.2442 *	0.1477 *	0.0744
IL-3	0.0336	0.0712	0.015	0.0214
IL-6	0.0081	0.0889	0.0023	0.122 *
IP-10	0.0022	0.1337 *	0.0002	0.0147
MIP-1β	0.0442	0.0087	0.0872	0.006
TNF-α	0.0016	0.0348	0.0103	0.1884 *

Values showed * weak correlation and ** moderate correlation.

## Data Availability

De-identified data for each study subject will be made available upon request and approval from all authors.

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
