# Peer review of "Plasma Cytokines/Chemokines as Predictive Biomarkers for Lymphedema in Breast Cancer Patients"

_cancers, 2023, doi:10.3390/cancers15030676_

Round 1
Reviewer 1 Report
This is a very interesting study. Could you do another study with prevention of BCRL with stockings CCL 1, because we know, that compression reduces pro-inflammatory cytokines and enhances the anti-inflammatory cytokines.
The main question addressed by the research is Early detection of lymphedema before it is visible. The topic is original and relevant in the field, as minimum of 20-30% (highest 40&) of BC patients will develop a lymphedema it is important for prevention. Compared with other published
material, this is a paper to detect the possibility of developing the lymphedema. The authors could consider specific improvements regarding the methodology of cytokines. The conclusions consistent with the evidence and arguments presented before and they addressed the main question posed.
Reviewer 2 Report
The authors provided an interesting and important clinical study, and the finding has significant clinical value. It will be better to add some comparisons of image results before and after the surgery. In addition, some inappropriate language usage needs to be corrected.
Reviewer 3 Report
This manuscript addresses an interesting topic that has been insufficiently exploited in the literature as a study
.Unfortunately-due to a large number of drop outs, the number of evaluable patients is considerably reduced and the statistical power of the study is weakened but this original study remains very interesting
I would ask to the authors to elaborate on the future perspectives:
-which breast cancer patients to offer cytokines screening to?
-What to do in case of positive screening?early management with physiotherapy? new therapeutic approaches? anti inflammatory drugs in post operative period? in post radiotherapy? duration? drugs?
